# APE: Aligning Pretrained Encoders to Quickly Learn Aligned Multimodal Representations

**Elan Rosenfeld**[*]
Carnegie Mellon University
elan@cmu.edu

**Preetum Nakkiran**
Apple

**Hadi Pouransari**
Apple

**Oncel Tuzel**
Apple

**Fartash Faghri**
Apple
fartash@apple.com

## Abstract

Recent advances in learning aligned multimodal representations have been primarily driven by training large neural networks on massive, noisy paired-modality datasets. In this work, we ask whether it is possible to achieve similar results with substantially less training time and data. We achieve this by taking advantage of existing pretrained unimodal encoders and careful curation of alignment data relevant to the downstream task of interest. We study a natural approach to aligning existing encoders via small auxiliary functions, and we find that this method is competitive with (or outperforms) state of the art in many settings while being less prone to overfitting, less costly to train, and more robust to distribution shift. With a properly chosen alignment distribution, our method surpasses prior state of the art for ImageNet zero-shot classification on public data while using two orders of magnitude less time and data and training 77% fewer parameters.

## 1 Introduction

How much modality-coupled data and compute is required to learn expressive, well-aligned multimodal representations? The latest advances in learning aligned representations have largely been driven by the compilation of ever-growing collections of noisy paired data scraped from the web (Radford et al., 2021; Jia et al., 2021). Trained on these massive multimodal datasets, new models achieve unparalleled performance on downstream tasks such as zero-shot classification, both in- and out-of-distribution (Radford et al., 2021; Hendrycks et al., 2019). Unfortunately, the cost of training these large models continues to scale in tandem—when little paired data already exists, or when one wants an aligned representation for a new setting, it is unclear how to avoid the time and expense of collecting and training on such a large dataset. Moreover, though it is simple to scale up noisy image-text pair scraping from the web, this is not necessarily the case for different modality couplings (e.g., audio descriptions of body pose) or more specific applications such as classification for niche downstream tasks.

In this work, we ask whether it is possible to leverage the power of pretrained unimodal encoders and a carefully chosen multimodal distribution to learn better aligned image-text representations with less training time and data. Our proposed approach, Aligning Pretrained Encoders (APE), results in well aligned, high-quality representations which can be learned orders of magnitude faster. We show

---

[*]Work done while an intern at Apple.

Has it Trained Yet? Workshop at the Conference on Neural Information Processing Systems (NeurIPS 2022).

that it is possible to align the representations of frozen pretrained encoders using simple functions with relatively few parameters (4-6 layer MLPs), substantially outperforming CLIP on zero-shot classification, with significantly less time spent aligning on multimodal data. Our method is inspired by Locked-image Tuning (LiT), which finetunes a text encoder to align with a frozen pretrained image encoder on a large paired-data corpus (Zhai et al., 2022). Instead, we consider settings with limited paired data, such as when the downstream task involves a distribution very different from the pretraining task or when we simply do not have the time and/or compute resources to train on all available pairs.

In this setting, we show how aligning pretrained encoders on a much smaller, carefully chosen dataset can result in better performance at less cost: our resulting model achieves 76.85% ImageNet zero-shot accuracy—as compared with 75.7% reported by LiT on public data—using 98% less training data and 98.5% less time on alignment. This suggests that collecting a small, high-quality dataset tailored to a specific downstream task can be significantly more cost- and compute-effective than scraping noisy data in bulk, in addition to providing better absolute performance. Further, we demonstrate that this simple approach is competitive even when training data is abundant, matching LiT to within 1.5% in-distribution and .5% under distribution shift while training approximately 20% as many parameters (Fig. 4).

**Related Work.**    The current state of the art in learning aligned multimodal representations is Contrastive Language-Image Pretraining (CLIP), which was demonstrated to be feasible at unprecedented scale by (Radford et al., 2021). Following this work, most advancements in this space have been primarily due to further scale in training set size (Jia et al., 2021), though a popular alternative is to simultaneously train the unimodal encoders on both unimodal and paired multimodal data (Geng et al., 2022). Zhai et al. (2022) demonstrate that using a *frozen, pretrained* image encoder results in substantially higher zero-shot accuracy on downstream classification tasks by making use of better visual representations. They train a large text encoder on the union of two public image-text datasets (with a total sample size of ~25 million) to align with a large ImageNet-21k pretrained Vision Transformer (Dosovitskiy et al., 2021). Though effective, the cost of training the text encoder remains, as well as the use of a massive amount of training data—their results are achieved by training for 60,000 iterations with a batch-size of 16,384. LiT is also prone to overfitting when the training set being used for alignment is not very large.

## 2    Method

To implement APE, we encode the paired data using separate pretrained unimodal image and text encoders and leave the image encoding unchanged. The token encodings of the text sample are passed through a small MLP (4-6 layers) and then average pooled across the sequence (See Fig. 3 for a high-level diagram). This does not directly account for token order; we instead rely on the output of the pretrained text encoder to include any relevant positional information.[2] The resulting embeddings are then normalized and used in the usual contrastive loss (Chen et al., 2020; Radford et al., 2021).

The MLP contains 7.5-22.5% the number of parameters in the entire text tower, which itself has slightly more parameters than the image encoder. More directly, LiT trains about half of the parameters trained by CLIP, and APE trains less than a quarter the number of parameters as LiT. Note that the total *number* of parameters is greater in APE, as we are learning a small MLP on top of the pretrained encoders—but APE is less likely to overfit to a small alignment dataset because it is training a much smaller fraction of these parameters. It is also cheaper to train because it avoids backpropagating through the large encoders, and some of the inputs can be pre-calculated to avoid having to load the encoders into memory at all. We found that text augmentations made little difference to final performance but image augmentations have a sizeable effect, so naively encoding all training data with the frozen encoder can result in sub-optimal downstream accuracy. Identifying the maximum reusable computation for various data modalities is an important future direction to investigate.

---

[2]Surprisingly, we found that training an auxiliary transformer actually performed worse than a simple MLP. To test that our method *is* making use of positional info in the text encoder output, we also tried directly learning a token embedding lookup table and average-pooling the results. It performs surprisingly well, but still much worse than APE (Fig. 4).

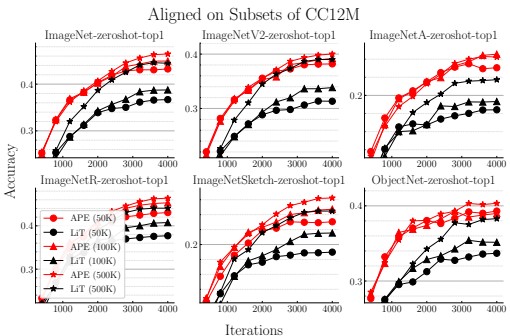
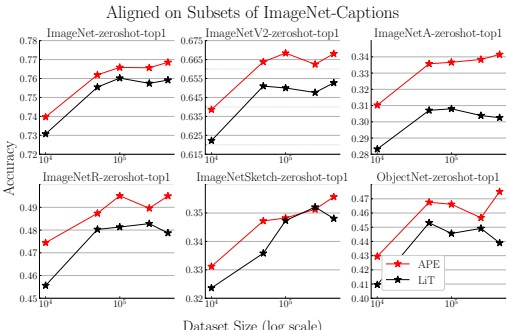

(a) 12K iterations on subsets of the large CC12M paired image-text dataset.

(b) 4K iterations on the much smaller ImageNet-Captions.

Figure 1: Comparison of APE to LiT when trained with (a) a relatively small amount of noisy paired data and (b) a high-quality dataset relevant to the downstream task.
**(a):** Parentheses give training subset size. With limited paired data, APE trains faster and reaches a higher accuracy than LiT both in-distribution (top left plot) and out-of-distribution (remaining plots, see Appendix C.1 for dataset details). **(b):** The benefits of APE are particularly apparent when training on carefully chosen alignment distributions specific to the downstream task. The best accuracy achieved by APE in this setting beats the previous SOTA set by LiT despite using two orders of magnitude less time and data.

**Additional benefits of small alignment functions.** Because the underlying encoders are frozen, it is easy to learn alignments for new downstream distributions or modalities (though we do not experiment with the latter). Currently, when encountering a new modality, it is unclear how to incorporate it without affecting the balance between existing aligned representations. Since APE does not modify the underlying pretrained text encoder, we can simply and cheaply align new representations without affecting existing alignment quality. This suggests a lightweight method for tying together new modalities as the need arises. Another advantage to keeping all encoders frozen is that we retain their powerful unimodal representations as needed. The text encoders of CLIP and LiT are primarily used for zero-shot classification, but they are not optimized for learning text representations; Zhai et al. (2022) observed that aligning an image encoder with a text encoder results in a worse unimodal image representations, and the reverse seems certain to hold (see Appendix B.1 for further discussion). By instead training a small auxiliary function, we get the best of both worlds by learning an alignment while retaining unimodal capabilities.

## 3 Experiments

We compare LiT to APE on Google's Conceptual Captions dataset (Changpinyo et al., 2021, CC12M), which consists of twelve million images with corresponding cleaned and partially anonymized captions (we observed similar qualititative results when training on YFCC (Thomee et al., 2016)). Note that we do not compare to CLIP, as CLIP takes requires much more time and memory to train and achieves substantially worse zero-shot accuracy in all settings (Zhai et al., 2022). See the Appendix for additional experiments plus details such as evaluation datasets and metrics, hyperparameters, and architectures.

To simulate a setting where massive amounts of paired data from the correct modalities are difficult to collect, we randomly subsample alignment sets of size 50K, 100K, and 500K from CC12M. Fig. 1a shows that in this setting with relatively little alignment data, APE trains faster and achieves a higher eventual accuracy than LiT. APE is particularly better under distribution shift; when the evaluation distribution is closer to the one on which the vision encoder was trained (i.e., ImageNet-21k to ImageNet-1k), the gap between the two methods shrinks. Consistent with the trend in Fig. 1a, we find that LiT does eventually outperform APE when using a massive amount of noisy paired data and wall-clock alignment time is sufficiently scaled up. However, the gap remains small, and given that APE is learning a simple weight-tied MLP on top of frozen token embeddings, it is quite surprising how close they are (see Fig. 4 in the Appendix).

**Collecting small amounts of high-quality data.** We next consider the setting where we have collected a small amount of "relevant" data—that is, paired data from a similar distribution to the

one we will be testing on. We use the recently introduced ImageNet-Captions dataset (INet-C) Fang et al. (2022). This dataset includes the original captions for ~446K images from the ImageNet train set (Deng et al., 2009) which are not typically included for the supervised learning benchmark. INet-C is approximately 25 times smaller than CC12M, but it is significantly more aligned with the task of zero-shot classification on ImageNet because the images are a subset of that dataset. We compare APE to LiT by using both methods to align on INet-C, as well as on random subsamples of sizes 250K, 100K, 50K, and 10K. We validate all models with image-text recall on the existing validation split of the smaller Conceptual Captions dataset. In addition to ImageNet zero-shot, we also evaluate on a standard suite of ImageNet variants to compare these methods' robustness to distribution shift. Fig. 1b shows that in this regime, APE consistently outperforms LiT, with a larger gap under distribution shift. This supports the idea that APE's lower parameter count allows it to better leverage small, high-quality datasets.

One additional takeaway is how much of a difference having the "right" data can make. Despite training on a much smaller dataset (as little as 10K image-text pairs!), the zero-shot accuracy remains high, and in fact APE aligned on the entirety of INet-C beats the state of the art set by LiT for zero-shot ImageNet accuracy trained on public data, at 76.85%. Further, **performing this alignment on a single 8-GPU machine took 1.75 hours—the best result reported for LiT requires aligning on almost a billion image-text pairs, which takes approximately 5 days of training using the same code and hardware.** This makes clear the enormous benefit of gathering high-quality data specialized for the desired task, and it suggests that even when it is entirely feasible to collect a large amount of noisy data, it may still be faster and cheaper to be selective.

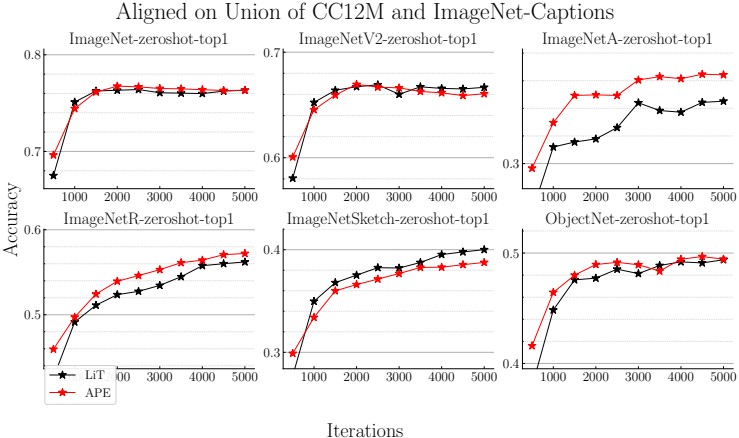

Figure 2: APE vs LiT on the union of CC12M and ImageNet-Captions. Zhai et al. (2022) train for 12K iterations on CC12M and reach lower zero-shot accuracy for all evaluations.

**Combining fewer, good data with more, noisy data.** Lastly, we consider the possibility that it may be most beneficial to simply *combine* all the paired data we have, since this is what originally enabled Zhai et al. (2022) to achieve such high downstream zero-shot accuracy. Fig. 2 compares APE to LiT when aligning on the union of the entirety of CC12M and INet-C. Recall that INet-C is just ~5% the size of CC12M, but by adding this small amount of data, both APE and LiT outperform the zero-shot accuracies reported by Zhai et al. (2022) while using less than half the alignment time. We note that here the gap between APE and LiT disappears for in-distribution evaluation, suggesting that when our *only* option is to train for a long time on lots of noisy data, LiT may still be preferable. However, the experiments presented here collectively make it clear that it is worth it to collect small amounts of the "right" data if one wishes to quickly learn an aligned multimodal representation.

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

# A    Diagram Contrasting Methods for Learning Aligned Multimodal Representations

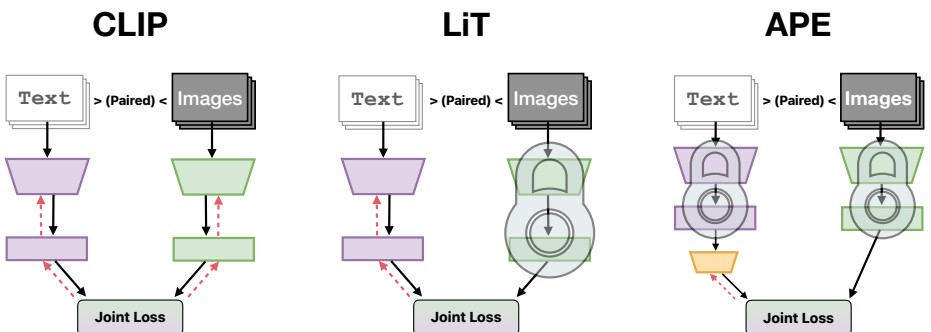

Figure 3: Diagram comparing CLIP, LiT, and APE. Using paired data, CLIP trains both image and text encoders from scratch, backpropagating the pairwise contrastive loss through both networks. LiT locks the image encoder at a pretrained initialization and trains the text encoder to align with it. Our method, APE, trains a much smaller MLP on the text representations, leaving *both* pretrained encoders untouched.

# B    Additional Experimental Results

## B.1    Training on the entirety of CC12M

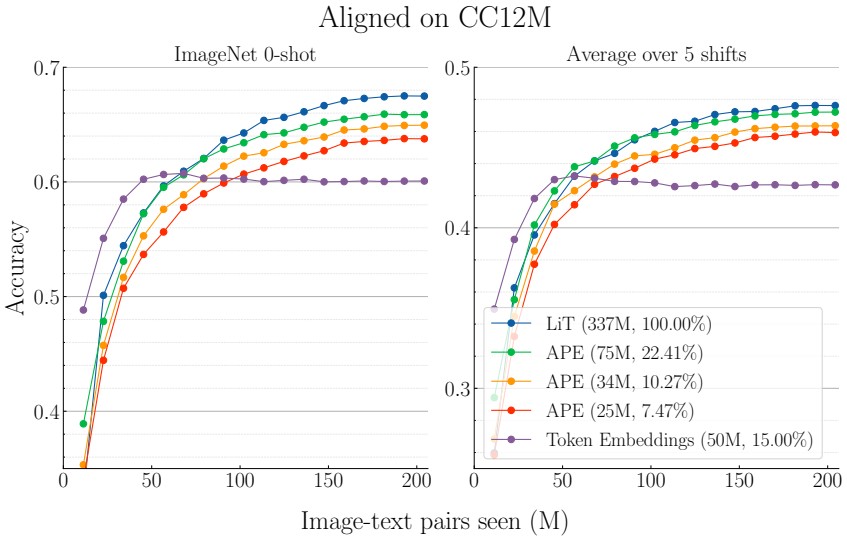

Figure 4: APE vs. LiT vs. raw token embeddings on the full CC12M paired dataset with a ViT-L/16 and BERT-L. Parentheses give number of parameters being trained and as a percentage of the number trained by LiT.

We consider the regime of abundant paired data, the setting in which LiT excels. Following Zhai et al. (2022), we train on CC12M for 200M seen pairs, amounting to about 12k iterations at a batch size of 16, 384, and we evaluate zero-shot classification accuracy on ImageNet and a suite of related distribution shifts. Consistent with findings that larger models perform better when trained on larger datasets, Fig. 4 shows that LiT does outperform APE in the setting (as a sanity check, our recreation of LiT outperforms the original implementation's reported results). However, given that APE is learning a simple weight-tied MLP on top of frozen token embeddings, it is quite surprising how close they are. Also, the fact that this gap shrinks under distribution shift suggests that the smaller

parameter count of APE may be beneficial when dealing with distribution shifts at test time, even with virtually unlimited alignment data and compute time.

Finally, as mentioned in the main body, we also evaluate the simple approach of learning a raw embedding for each vocabulary token and simply averaging the encodings of all tokens in a sequence. Though this approach is still surpassed by APE and LiT by a large margin, it performs surprisingly well, giving further evidence to the idea that a powerful text encoder is not necessary in order to perform well at zero-shot classification. This implies that the text encoder learned by CLIP and LiT may not be suitable for unimodal text-based downstream tasks, which is why freezing the text encoder as done by APE is even more beneficial.

## B.2 Training on additional mixtures of CC12M and ImageNet-Captions

To complement Fig. 2, in Fig. 5 and Fig. 6 we plot the results of training APE on LiT on other mixtures of CC12M and ImageNet-Captions. All mixtures include the entire 446K samples from ImageNet-Captions, plus varying size subsets of CC12M. The legend displays the ratio of CC12M samples to ImageNet-Captions samples—i.e., 2:1 means the alignment dataset is approximately 1.5M samples, 500K from INet-C and 1M from CC12M. The ratio of 51:2 represents the full mixture, as presented in Fig. 2. The value in parentheses is the total alignment dataset size.

We see the same consistent pattern, with APE outperforming LiT in all settings when paired data is limited, with the gap shrinking as the alignmented dataset grows. Also, the gap grows larger under more substantial distribution shift such as zero-shot classification on ImageNet-A. This raises interesting questions about the qualitative manner in which various distributions are shifted from one another and how APE or LiT may perform better under different *kinds* of shifts depending on which data was used for pretraining and/or encoder alignment.

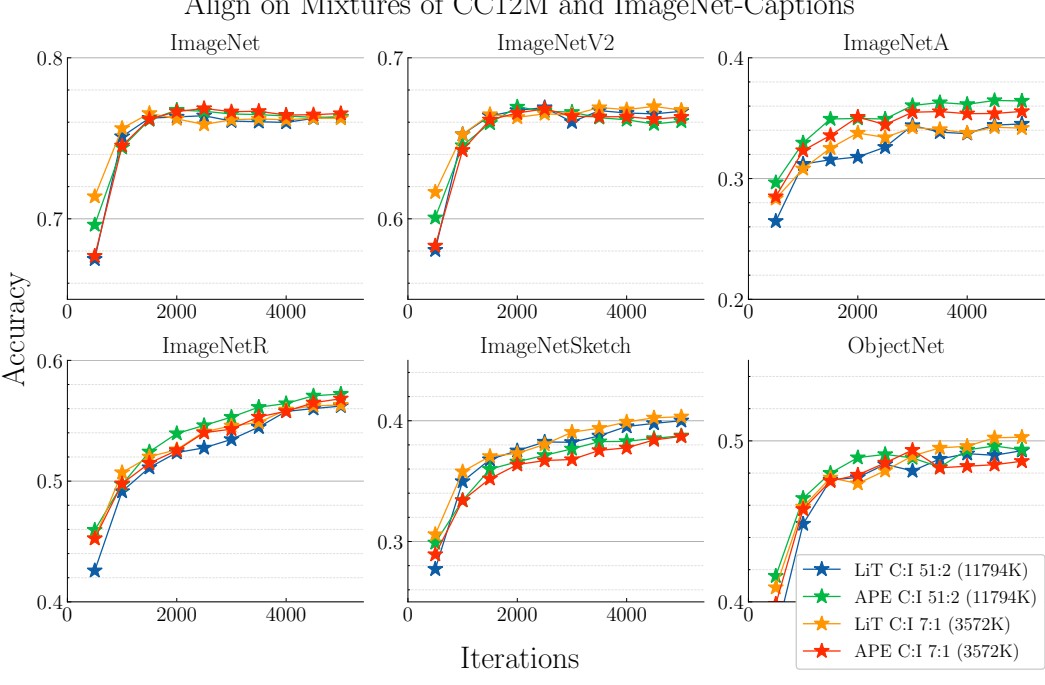

Figure 5: APE vs LiT on mixtures of CC12M and INet-C, where the high-quality INet-C samples are heavily dominated by the noisy CC12M samples.

## B.3 Learning an MLP on top of the image encoder

Fig. 7 displays the effect of also training an MLP on top of the frozen image encoder. Like Zhai et al. (2022), we find that modifying the image representation to try to improve alignment results in worse image features overall, harming downstream accuracy. Thus it seems clear that it is better to leave the

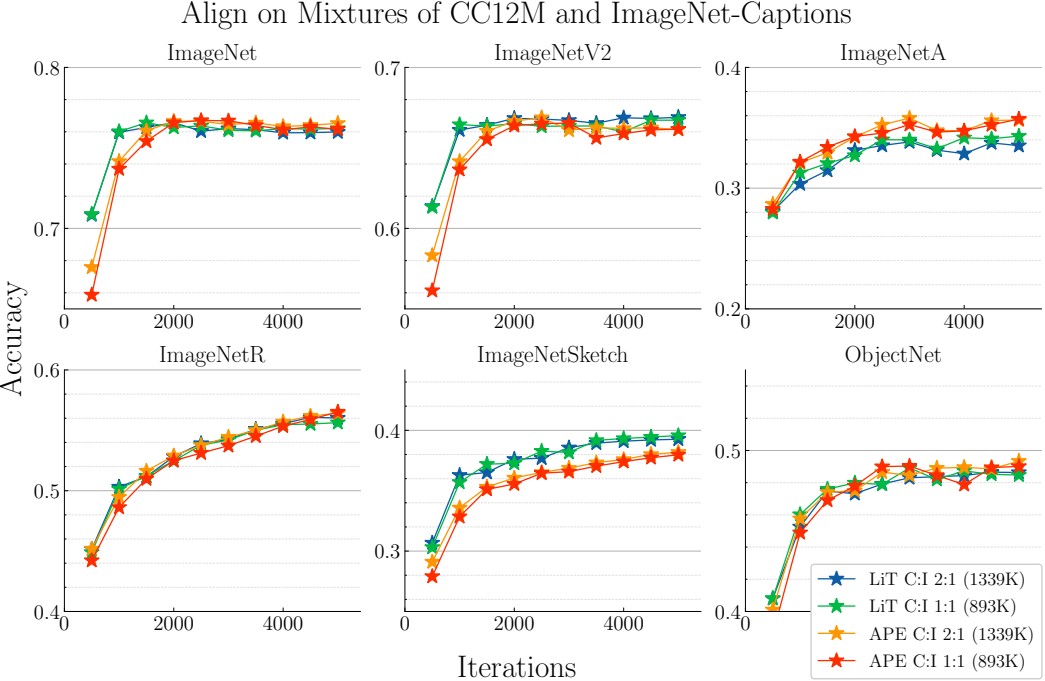

Figure 6: APE vs LiT on mixtures of CC12M and INet-C, where the high-quality INet-C samples comprise a reasonably large fraction of the entire alignment set and contribute more to the overall distribution.

representations more important for a given downstream task unmodified (e.g., for a text-focused task we would prefer to leave the text representation untouched and align only the image encoder).

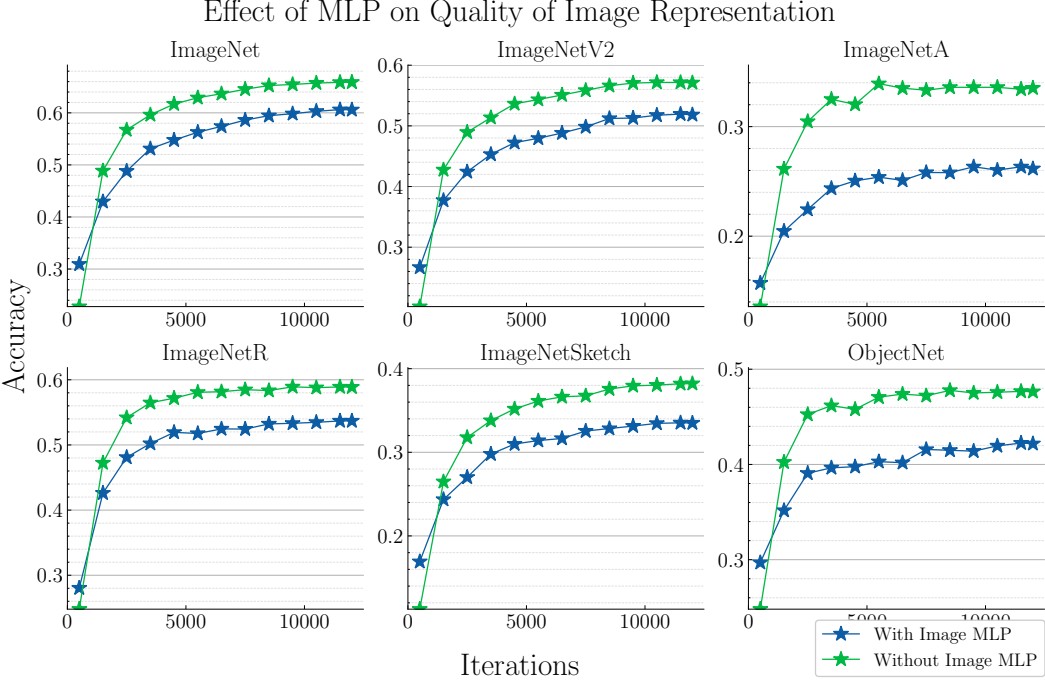

Figure 7: APE with and without an MLP trained on top of the image representation. Like Zhai et al. (2022), we find that modifying the image representation to try to improve alignment results in worse image features overall, harming downstream accuracy.

# C  Experimental Details

## C.1  Datasets

We train primarily on CC12M (except where indicated otherwise) and validate on the smaller Conceptual Captions dataset (Sharma et al., 2018, CC3M). Where directly recreating LiT results we use the hyperparameters reported in that paper.

We evaluate in-distribution ImageNet zero-shot classification using class templates as described by Radford et al. (2021). Out-of-distribution evaluation is on a standard suite of ImageNet distribution shifts: ImageNetV2 (Recht et al., 2019), ImageNet-A (Hendrycks et al., 2021b), ImageNet-R (Hendrycks et al., 2021a), ImageNet-Sketch (Wang et al., 2019), and ObjectNet (Barbu et al., 2019).

Our code is built on top of the open-source implementation of CLIP provided by Ilharco et al. (2021).

## C.2  Architecture and Hyperparameters

For all experiments in the paper the image encoder is an ImageNet-21k supervised pretrained ViT-L/16 with the suggested checkpoint from Steiner et al. (2022). The text model is a pretrained BERT model (Devlin et al., 2019), either bert-base or bert-large with the final layer removed, providing encodings for each token in the embedded sequence. In reimplementing LiT we found training with bert-large to be very unstable, frequently collapsing and requiring training restarts, so we use bert-base except for when we use the full training set in Fig. 4. Like Zhai et al. (2022) we found that when bert-large did converge it was to similar downstream performance—the size of the image encoder was significantly more important.

All methods use a linear learning rate warmup followed by a cosine decay and the Adam optimizer with decoupled weight decay (Loshchilov & Hutter, 2017). We select learning rate, weight decay, and warmup duration by validation of image-to-caption recall accuracy on CC3M. Following Zhai et al. (2022) we use a batch size of $2^{14}$ on the full CC12M, and smaller batch sizes ranging from $2^9$ to $2^{12}$ for smaller training sets.

