# OpenReview forum: "APE: Aligning Pretrained Encoders to Quickly Learn Aligned Multimodal Representations"
_NeurIPS.cc/2022/Workshop/HITY — HITY Workshop NeurIPS 2022_

### Official Review · Reviewer_RUx9 · 2022-10-17

**Rating:** 1
**Confidence:** 4

**Review:**

This work presents a new approach for efficiently training multi-modal (image-text) models by aligning existing pre-trained unimodal encoders. The paper is very clear and the results are impressive, showing improvements over state-of-the-art models such as LiT (let alone traditional multi-modal training methods such as CLIP) on the challenging zero-shot ImageNet classification task.

---

### Official Review · Reviewer_qbyu · 2022-10-17
**Accept: Interesting results on using pretrained model components and curated data for learning aligned multimodal representations**

**Rating:** 1
**Confidence:** 3

**Review:**

The paper proposes to reduce the training time and data for learning aligned multimodal representations by leveraging pretrained unimodal encoders and curation of the training data. This results in significantly faster training time, due to using less data and training fewer parameters.

I recommend to accept the paper, since it provides interesting evidence for the power of using pretrained components within a larger multimodal model and on the importance of data curation, even when large amounts of noisy data are available.

---

### Decision · Program_Chairs · 2022-10-20

Accept